# High-Resolution Genomic Profiling of a Genotype 3b Hepatitis C Virus from a Flare of an Occult Hepatitis Patient with Acute-on-Chronic Liver Failure

**DOI:** 10.3390/v15030634

**Published:** 2023-02-26

**Authors:** Xue Mei, Jingyi Zou, Bisheng Shi, Zhiping Qian, Zhigang Yi

**Affiliations:** 1Shanghai Public Health Clinical Center, Fudan University, Shanghai 201508, China; 2Key Laboratory of Medical Molecular Virology (MOE/NHC/CAMS), School of Basic Medical Sciences, Shanghai Institute of Infectious Disease and Biosecurity, Fudan University, Shanghai 200032, China

**Keywords:** hepatitis C virus, genotype 3b, occult hepatitis, acute-on-chronic liver failure

## Abstract

Acute-on-chronic liver failure (ACLF) is defined as a syndrome of acutely decompensated cirrhosis in patients with chronic liver disease (CLD). Here we report an ACLF case caused by a flare of occult hepatitis C infection. This patient was infected with hepatitis C virus (HCV) more than a decade ago and hospitalized due to alcohol-associated CLD. Upon admission, the HCV RNA in the serum was negative and the anti-HCV antibody was positive, whereas the viral RNA in the plasma dramatically increased during hospitalization, which suggests an occult hepatitis C infection. Overlapped fragments encompassing the nearly whole HCV viral genome were amplified, cloned, and sequenced. Phylogenetic analysis indicated an HCV genotype 3b strain. Sanger sequencing to 10-fold coverage of the 9.4-kb nearly whole genome reveals high diversity of viral quasispecies, an indicator of chronic infection. Inherent resistance-associated substitutions (RASs) in the NS3 and NS5A but not in the NS5B regions were identified. The patient developed liver failure and accepted liver transplantation, followed by direct-acting antiviral (DAA) treatment. The hepatitis C was cured by the DAA treatment despite the existence of RASs. Thus, care should be taken for occult hepatitis C in patients with alcoholic cirrhosis. The analysis of viral genetic diversity may help to identify an occult hepatitis C virus infection and predict the efficacy of anti-viral treatment.

## 1. Introduction

Hepatitis C virus (HCV) is a member of the *hepacivirus* genus within the *Flaviviridae* family. HCV chronically infects approximately 71 million people worldwide [1]. Approximately 80% of HCV-infected patients develop chronic infection [2]. HCV infection causes hepatocellular carcinoma (HCC) in a significant proportion of the chronically infected population. Viral infection-associated HCC represents the leading cause of death from cancer worldwide [3].

HCV has a 9.6-kb positive-sense RNA genome, which encodes a polyprotein that is co- and post-translationally processed to produce the viral structural proteins C, E1, and E2 and the nonstructural proteins NS2, NS3, NS4A, NS4B, NS5A, and NS5B [4]. The nonstructural proteins NS3, NS4A, NS4B, NS5A, and NS5B assemble the viral replicase for replication. NS3 contains a protease domain in its N-terminal portion. NS4A acts as the cofactor for NS3 protease. NS5A is a multifunctional viral protein with non-enzymatic activity and plays essential roles in viral replicase assembly. NS5B is the viral RNA-dependent-RNA polymerase [5]. Direct-acting antivirals (DAAs) against the NS3 protease, NS5A, and NS5B were developed, and treatments with combinations of DAA cocktails cure more than 90% of chronically HCV-infected patients [6].

Occult hepatitis C virus (HCV) infection (OCI) is defined as undetectable HCV RNA in the serum while it is present in the liver [7,8]. The detection of HCV RNA in liver biopsy is the gold standard for the diagnosis of OCI. However, in certain circumstances, liver biopsy is not applicable, such as for patients with severe liver disease. It was reported that in up to 70% of OCI patients, viral RNA has been detected in the peripheral blood mononuclear cells (PBMCs) [7]. The detection of HCV-RNA in PBMCs and in concentrated serum samples, combined with the detection of anti-core HCV, has been used as an alternative method for the diagnosis of OCI [9].

Acute-on-chronic liver failure (ACLF) is described as a syndrome in patients with acutely decompensated cirrhosis [10]. ACLF is associated with a high risk of short-term death. It usually develops in the context of systematic inflammation. Flares of chronic hepatitis B, acute hepatitis E, and hepatitis A have been implicated as precipitating factors for ACLF [11,12].

In this study, we report an ACLF case with a flare of occult hepatitis C virus infection. This patient was infected with HCV more than a decade ago. The anti-HCV antibody was positive, but HCV RNA was negative upon hospital admission, while during hospitalization, HCV RNA in the plasma burst. Genotyping the viral sequence indicated an HCV genotype 3b infection. High-resolution sequencing of the viral genome revealed a high diversity of viral quasispecies, an indicator of chronic infection [13]. The patient was eventually subjected to liver transplantation, followed by DAA treatment, and the hepatitis C was cured despite the existence of multiple inherent resistance-associated substitutions in the viral genome. Thus, care should be taken for occult hepatitis C in patients with alcoholic cirrhosis, and analyzing the viral genetic diversity may help to identify an occult hepatitis C virus infection.

## 2. Materials and Methods

### 2.1. Ethics

This study was reviewed and approved by the Ethics Committee of Shanghai public health clinical center, Shanghai Medical College, Fudan University. The procedures were carried out in accordance with the approved guidelines. Informed consent was obtained from the subject.

### 2.2. RNA Extraction and Reverse Transcription

RNA from plasma was extracted by a QIAamp^®^ Viral RNA Mini kit (Qiagen, Beijing, China) and eluted in RNase-free H_2_O according to the manufacturer’s protocol. Reverse transcription was performed with Superscript IV (Invitrogen, Beijing, China). Briefly, 8 μL of RNA samples was reversely transcribed with 1 μL of random primer (dN6) or gene-specific primers according to the manufacturer’s protocol.

### 2.3. Quantification of HCV RNA by Real-Time PCR (R-PCR)

For clinical diagnosis, HCV RNA in the plasma was analyzed by the Sansure Biotech (Hunan, China) HCV Ultra—Hepatitis C Virus RNA Quantitative Fluorescence Diagnostic Kit, with a verified detection limit of 1000 IU/mL, or by the Roche COBAS Ampliprep/COBAS TaqMan HCV Test (CAP/CTM), with a detection limit of 15 IU/mL.

For the laboratory experiments, the HCV RNA was quantified by real-time PCR with a TaqMan-based method. Briefly, 2.5 μL of RNA was reversely transcribed and PCR-amplified by the TAKARA (Beijing, China) One Step PrimeScript RT-PCR Kit in 25 μL of reaction mixture, as follows: 2× one-step RT-PCR buffer, 12.5 μL; EX Taq HS, 0.5 μL; RT enzyme, 0.5 μL; primer mix (10 μM each) 1.6 μL; probe (10 μM), 0.4 μL; H_2_O up to 25 μL with an R-PCR program (42 °C, 10 min; 95 °C, 10 s; 95 °C 10 s, 60 °C, 1 min) for 40 cycles. The forward primer (HCV-Con259F-AGY GTT GGG TYG CGA AAG), reverse primer (HCV-Con312R-CAC TCG CAA GCR CCC T), and probe (HCV-278MGB-6FAM CCT TGT GGT ACT GCC TGA) were used as described previously [14].

### 2.4. HCV Genotyping

For HCV genotyping, cDNA was used as a template for PCR amplification with superFi DNA polymerase (Invitrogen, Beijing, China) in 25 μL of reaction mixture as follows: 5× SuperFi buffer, 5 μL; GC enhancer, 5 μL; 10 mM dNTP mix, 0.5 μL; primer mix (10 μM each), 2.5 μL; template cDNA, 1.5 μL; SuperFi polymerase, 0.5 μL. The primer sequences were sense primer (ACT GCC TGA TAG GGY GCT TGC) and antisense primer (ATG TAC CCC ATG AGR TCG GC) as described previously [15]. PCR amplification was carried out using the following program: 98 °C for 30 s; 98 °C for 10 s, 58°C for 10 s, 72 °C for 1 min, 40 cycles; 72 °C 1 for 0 min. The 405-nt PCR products were gel-purified and ligated into a homemade pZero-Blunt cloning vector. Colonies were propagated, and plasmid was prepared and sequenced by Sanger sequencing with primers on the vector backbone.

### 2.5. Cloning and Sequencing the Whole HCV Genome

For cloning the whole HCV genome, 2 μL of cDNA was used as a template for PCR amplification with superFi DNA polymerase in a 25 μL reaction mixture as described above, with primers for 6 fragments encompassing the nearly whole viral genome (Table 1). PCR amplification was carried out using the following program: 98 °C for 30 s; 98 °C for 10 s, 58 °C for 10 s, 72 °C for 1 min, 40 cycles; 72 °C for 10 min. The PCR products were gel-purified and ligated into a pZero-Blunt vector. For each cloned fragment, at least 10 colonies were propagated and sequenced by Sanger sequencing, with primers on the vector backbone. For sequencing certain inserts, additional internal sequencing primers were designed and used based on the sequenced sequences.

### 2.6. Bioinformatics

The DNA sequences were aligned and the contigs were assembled by Seqman (DNAstar, Lasergene, Madison, WI, USA). The DNA sequences were analyzed and translated by Editseq (DNAstar, Lasergene, Madison, WI, USA). Molecular phylogenetic analysis was conducted using the maximum likelihood method. The evolutionary history was inferred using the maximum likelihood method based on the Tamura–Nei model [16]. The tree with the highest log likelihood (−20,659.81) is shown. The initial tree(s) for the heuristic search were obtained automatically by applying the neighbor-join and BioNJ algorithms to a matrix of pairwise distances estimated using the maximum composite likelihood (MCL) approach, and then selecting the topology with the most superior log likelihood value. The tree is drawn to scale, with branch lengths measured in the number of substitutions per site. Evolutionary analyses were conducted with MEGA7, accessed on 1 January 2021 (http://megasoftware.net (accessed on 1 January 2021)) [17].

## 3. Results

### 3.1. Clinical Features of the Patient

A 39-year-old man with about a 10-year history of heavy drinking was admitted to the hospital on 4 June 2018 due to fatigue, poor appetite, and abdominal distension for a week. Physical examination upon admission found that the skin and sclera were severely yellow-stained, the liver palms and spider nevi were positive, shifting dullness was positive, and the lower limbs had edema. He was diagnosed with decompensation of alcoholic cirrhosis and ascites.

Upon admission, Doppler ultrasound confirmed liver cirrhosis, fatty liver, widened portal vein, splenomegaly, and massive peritoneal effusion. The laboratory test showed that the hemoglobin (HB) was 14.1 g dL, the platelet (PLT) count was 26 × 10^9^/L, and the international standardized ratio (INR) was 1.93. Liver function assays indicated that the total bilirubin (TB) was 306 μmol/mL and the albumin (Ab) was 30 g/L (Figure 1). Screening for hepatitis viruses indicated that HBsAg and HBV DNA were negative, HBsAb and HBcAb were positive, and anti-HEV and anti-HAV IgM antibodies were negative. A panel of autoimmune disease-related antibodies tested negative. A CT of the chest and abdomen showed cirrhosis, splenomegaly, and ascites, but there was no evidence of malignancy. After admission, the patient’s liver function deteriorated due to biliary tract infection and ascites. The levels of TB increased to 878 μmol/L, and the INR increased to 2.15 on 27 June 2018. After the infection and ascites were controlled, the patient’s liver function gradually improved. However, during hospitalization, the patient repeatedly suffered from gastrointestinal tract bacterial infections and ascites. The patient had lower than normal CD3, CD4, and CD8 T cell counts, and lower lipoproteins HDL and LDL in the blood, indicating a disorder of the lipoprotein metabolism (Table 2).

Eventually, on 2 November 2018, he was diagnosed with acute-on-chronic liver failure (ACLF) [18], and his platelets and hemoglobin continued to decline. At the same time, his mobility decreased, his nutrition deteriorated, and his muscle cells decreased. Although he received spironolactone, furosemide diuresis, albumin infusion, and intravenous antibiotics, the model for end-stage liver disease (MELD) score [19,20] continued to increase to 29 points on 9 November 2018, and he was finally placed on the waiting list for liver transplantation (Figure 1).

### 3.2. Diagnosis of Hepatitis C Infection

The patient explained that he had been infected with hepatitis C more than a decade previous and was anti-HCV antibody-positive, but he had normal liver function and did not receive any antiviral treatment. When he was admitted to this hospital, the anti-HCV Ab was 14.15 (s/co), but the viral RNA level in the serum was lower than the limit of detection (1000 IU/mL). After admission, the patient took another two HCV RNA tests, in which the viral RNA was not detected in the serum by a highly sensitive accurate quantification assay (Roche COBAS Ampliprep/COBAS TaqMan virus load detection system) with a detection limit of 15 IU/mL. Thus, the patient did not receive direct-acting antiviral (DAA) treatment. However, three months later (on 7 October 2018), the patient’s liver function deteriorated, and the MELD score rose progressively to 29. We retested the HCV RNA in the serum, and the viral RNA was 34,000,000 IU/mL, indicating active viral replication (Figure 1). The patient’s wife and other family members had no history of hepatitis C virus infection, and the other patient in his inpatient ward was a patient with hepatitis B cirrhosis but who was anti-HCV antibody-negative, which excludes the possibility of a new HCV infection in this case during hospitalization. These data imply that this case was occult hepatitis C.

The patient was diagnosed with acute-on-chronic liver failure (ACLF) on 2 November 2018 and it was suggested that he undergo liver transplantation. Finally, he received liver transplantation on 20 November 2018. After the operation, he received DAA treatment (Epclusa—sofosbuvir 400 mg/velpatasvir 100 mg). Three weeks later, the viral RNA could not be detected in the serum (Figure 1). In the subsequent follow-ups (longer than 24 weeks after treatment), the HCV RNA level in the serum was lower than the limit of detection. The liver function was normal and the liver ultrasound showed no obvious fibrosis change (data not shown). Thus, the hepatitis C was cured after liver transplantation.

### 3.3. Cloning the Genome of HCV and Phylogenetic Analysis of the HCV

HCV viral RNA from the plasma was extracted and reversely transcribed. First, we verified the presence of viral RNA by a TaqMan assay as described previously [14] (data not shown). Then, we genotyped the HCV as previously reported [15]; a 405-nt fragment encompassing part of the 5′-non-translated region (5′-NTR) and the core region was amplified, cloned, and sequenced. The viral sequence was blasted as HCV genotype 3b.

Based on the sequences of genotype 3b in the database, we designed primers to amplify seven overlapped fragments encompassing the whole genome. Except for F7 containing the 3′-non-translated region (3′-NTR), six fragments (F1–6) encompassing the nearly whole viral genome were successfully amplified. The amplicons were cloned and sequenced. HCV exists in an infected individual as a population of quasispecies with genetic variations, and the genetic diversity indicates the virus–host interaction under the host’s immune pressure during viral persistence [21]. To capture the viral genetic heterogeneity, for each of the cloned fragments, we sequenced at least 10 colonies with the vector-derived sequencing primers using the Sanger sequencing method.

A 9.4-kb nearly whole-genome sequence was obtained and named HCV-3b-js1. The consensus sequence of the HCV-3b-js1 was blasted with the HCV sequences in the database. Similar sequences identified were extracted and used for generating a phylogenetic tree using the MEGA 7 program, as described in Section 2. The phylogenetic analysis indicated that it was an HCV 3b strain with a sequence close to that of a genotype 3b strain that circulated in China around 2013 (Figure 2, in red).

### 3.4. Sequence Profile of HCV Genome

Sequencing the viral genome at 10-fold coverage allowed for profiling the viral genome at a high resolution. The sequenced colonies of the cloned fragments revealed high genetic heterogeneity of the viral sequence. There were 5 different sequence patterns out of the 10 sequenced colonies in F1 that contained part of the C-E1 region; 10 different sequence patterns out of the 11 sequenced colonies in F2 contained part of the E1-E2-P7 region; 10 different sequence patterns out of the 10 sequenced colonies in F3 contained part of the P7-NS2-NS3 region (Figure 3); 10 different sequence patterns out of the 10 sequenced colonies in F4 contained part of the NS3-4A-4B-5A region; 8 different sequence patterns out of the 10 sequenced colonies in F5 contained part of the NS5A-5B region; and 2 different sequence patterns out of the 10 sequenced colonies in F6 contained part of the NS5B region (Figure 4). Within each fragment, the sequenced colonies shared certain variation patterns, e.g., clone-3 and clone-4 in F2; clone-2 and clone-3 in F3; clone-1 and clone-4 in F4; and clone-3, clone-4, clone-5, and clone-6 in F5 (Figure 3 and Figure 4). These shared variation patterns may indicate the evolution routes of the viral quasispecies.

The viral sequence variations were unevenly distributed across the structural protein region and the non-structural protein region, with lower variation rates in the C-E1 and part of the NS5B region (Figure 3 and Figure 4). The higher frequency of quasispecies in the E2 region (10/11) than in the C-E1 region (5/10) may indicate that E2 encountered greater pressure from the host immune system (Figure 3 and Figure 4). Moreover, diverse variations distributed within the viral non-structural protein regions (Figure 3, Figure 4 and Figure 5A) may indicate a long-term virus–host interaction during viral persistence. Of the variations within the viral non-structural protein regions, there were hotspot variations with a ratio above 3/10 that existed in the NS5A DII-DIII (domain II–domain III) linker region (P2325S), the NS5A DIII (domain III) region (M2341I, T2423S), and the NS5B region (N2497D/S) (Figure 5B). The NS5A P2325 region is within a conserved region that is required for viral replication [22].

### 3.5. Resistance-Associated Substitutions (RASs) in the Viral Genome

HCV genotype 3 has been shown to be less sensitive to direct-acting antiviral treatment [23,24]. There are inherently resistance-associated substitutions (RASs) to NS5A inhibitors in genotype 3-infected, treatment-naïve patients [25]. We analyzed the viral genomes of HCV 3b-js1 and aligned the genome with the reported RASs [26] to identify the potential inherent RASs in the NS3, NS5A, and NS5B regions. There were intrinsic RASs in the HCV 3b-js1 NS3 (D168Q, V170I) and NS5A (Q30K, L31M, H58P) regions but not in the NS5B regions (Figure 6). NS3.D168A, NS3.D168E, NS3.D168V, NS3.D168Y, NS3.D168H, NS3.D168I, NS3.D168K, NS3.D168N, NS3.D168T, and NS3.V170A were identified as RASs [26]. Whether the D168Q and V170I identified in this study are also RASs needs to be verified. As NS5A.Q30R is a reported RAS [26], and a similar substitution, Q30K, identified in this study, is likely an RAS. Given that NS5A.H58D is a reported RAS [26], whether the H58P identified in this study is an RAS needs to be verified.

## 4. Discussion

Acute-on-chronic liver failure (ACLF) development is frequently associated with proinflammatory events such as alcoholic hepatitis or viral infection [11,12]. This patient was infected with hepatitis C more than a decade ago, as evidenced by testing anti-HCV antibody-positive, despite the HCV RNA diagnosis not being available at the time. The patient did not receive any treatment at that time because the liver function was normal. When he was admitted to this hospital, the anti-HCV Ab was 14.15 (s/co), but a highly sensitive and accurate quantification assay using the Roche COBAS kit showed that the HCV RNA in the serum was negative (Figure 1), which suggested occult hepatitis C infection. The diagnosis of occult hepatitis C virus infection (OCI) relies on detecting HCV RNA in liver biopsy when the HCV RNA in the serum is negative. However, liver biopsy is not always applicable to patients with severe liver disease. Alternative methods, such as a combination of viral RNA detection in PBMCs and concentrated serum and the detection of anti-HCV antibodies, have been used for the diagnosis of OCI [9].

Three months after admission, we detected a burst of the HCV RNA level in the plasma of the patient, suggesting a flare of HCV (Figure 1). In a previous study, analysis of the HCV quasispecies following transmission in patients infected by blood transfusion showed that the quasispecies composition remained static in acutely infected individuals, whereas, in chronically infected patients, the quasispecies underwent changes [13], demonstrating that the diversity of viral quasispecies is associated with chronicity of viral infection. In order to verify if the burst of the HCV RNA level in the plasma in this patient was due to the flare of an occult hepatitis C virus infection or a new acute infection, we performed a high-resolution analysis of the viral sequences using Sanger sequencing. Diverse genetic variations were readily identified within almost the whole viral genome (Figure 3, Figure 4 and Figure 5), indicating that this HCV infection was unlikely a new acute infection, but a flare of occult hepatitis C virus infection. Taking these data, we deduced that this patient might have been infected with occult hepatitis C infection for the first time more than a decade ago.

This patient had about a 10-year history of heavy drinking and was admitted to the hospital due to alcoholic hepatitis. The development of ACLF in this case might have been due to the combination of alcoholic hepatitis and the flare of long-term occult hepatitis C infection. In patients with alcoholic hepatitis, ACLF is a frequent and severe complication, characterized by immune dysfunction and associated with an increased risk of infection [27]. We think that the flare of HCV in this patient was probably due to the dysfunction of the immune system, which is important for controlling HCV infection. Indeed, we observed lower CD4 and CD8 T cell counts in this patient (Table 2).

This patient was infected by an HCV genotype 3 (Figure 2). It is reported that the annual average reported incidence rate of acute hepatitis C in Shanghai from 2014 to 2019 was 0.18/100,000, of which type 3b accounted for 14.92% [28]. HCV genotype 3 is a difficult-to-treat genotype [23,24], probably due to the presence of inherently resistance-associated substitutions (RASs) in the viral genome [25]. In this case, we found several intrinsic RASs in the viral genome, such as the NS5A. L31M and NS5A.Q30K. Moreover, there were several potential RASs with the same amino acid positions as the reported RASs but with different substitutions, such as substitutions in NS3 (D168Q, V170I) and NS5A (H58P). No reported RASs were detected in the NS5B region (Figure 6). This patient eventually received liver transplantation, and the HCV infection was cured by DAA treatment (Epclusa—sofosbuvir plus velpatasvir), a combination of NS5B and NS5A inhibitors, even though RASs were present in the viral genome, which was probably due to the absence of RASs in the NS5B region and the inclusion of NS5B inhibitor sofosbuvir in the treatment. Moreover, the low viral load after the liver transplantation (6580 IU/mL) (Figure 1) indicates a low diversity of viral quasispecies, making the virus sensitive to antiviral treatment.

This study has limitations. In this study, although the HCV RNA in the serum of the patient was negative upon admission to the hospital, we cannot exclude the possibility of the presence of viral RNA in the blood. The concentration of the virus in the blood might help to detect viral RNA. Due to the availability of the samples, we could not directly confirm the presence of viral RNA in the serum or liver biopsy when this patient was infected at the first time point, which was a decade ago.

In summary, we report an acute-on-chronic liver failure patient with a flare of occult hepatitis C. The diverse quasispecies identified in the viral genome confirmed long-term occult hepatitis. The presence of inherently resistance-associated substitutions was identified in the viral NS3 and NS5A but not the NS5B. The HCV infection was cured after liver transplantation, followed by DAA treatment with a combination of NS5B and NS5A inhibitors. Our study emphasizes that care should be taken for occult hepatitis C, especially in patients with liver diseases such as alcoholic hepatitis. Analysis of the viral genetic diversity in occult hepatitis C may help to identify occult hepatitis C virus infection and predict the efficacy of anti-viral treatment.

## Figures and Tables

**Figure 1 viruses-15-00634-f001:**
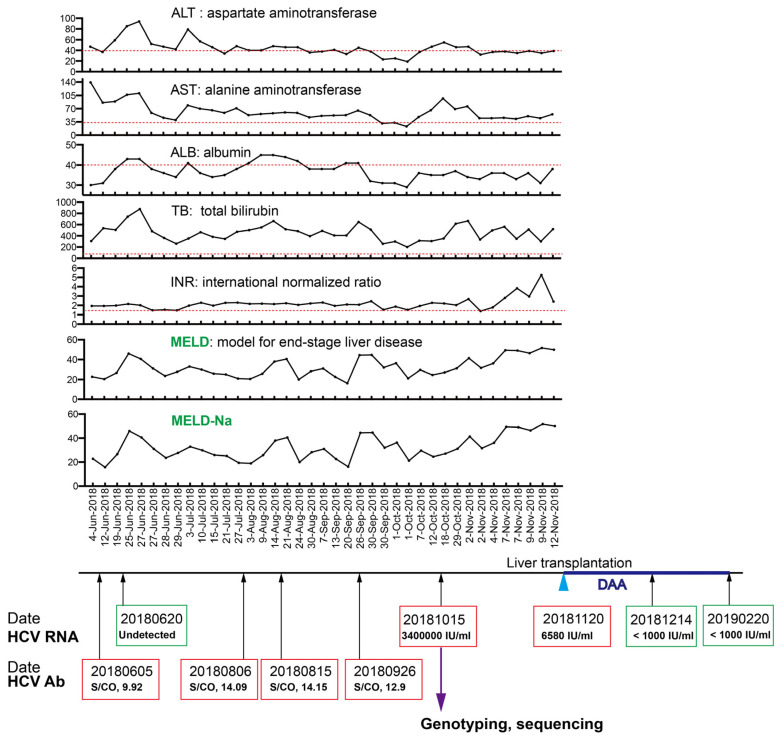
Clinical features of the hepatitis C patients. The aspartate aminotransferase (AST, upper limit of normal = 40 U/L), alanine aminotransferase (ALT, upper limit of normal = 35 U/L), albumin 9ALB, upper limit of normal = 35 U/L), total bilirubin (TB, upper limit of normal = 17.1 μmol/L), international normalized ratio (INR, upper limit of normal = 1.5), model for end-stage liver disease (MELD) and the MELD-Na at each time point are presented. Red dashed lines indicate the upper limits of normal. Plasma viral RNAs were quantified at the indicated time points. Direct antiviral agent (DAA) treatment is shown in blue. Plasma samples (purple arrow) were applied for viral genotyping and sequencing. See details in the main text.

**Figure 2 viruses-15-00634-f002:**
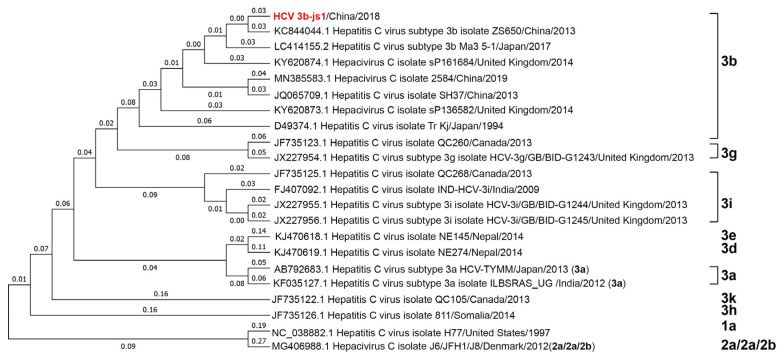
Phylogenetic analyses of HCV3b-js1. The consensus sequence of the HCV-3b-js1 (in red) was blasted and the phylogenetic tree was generated with Mega7, as described in Section 2.

**Figure 3 viruses-15-00634-f003:**
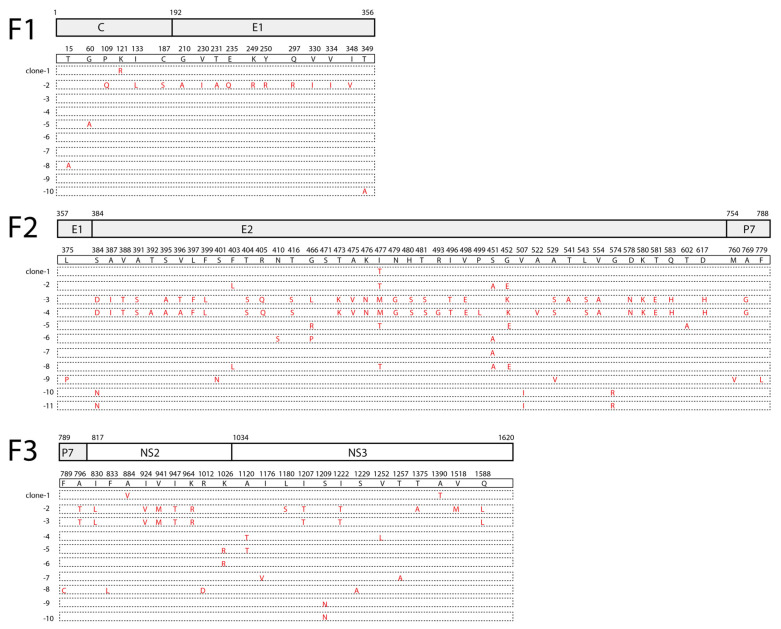
Sequence profile of HCV genome fragments−I. As described above, for each amplified fragment, the fragment was cloned, and at least ten colonies were sequenced. Each dashed box indicates a sequenced cloned fragment. The amino acids different from the consensus sequence in each sequenced clone are indicated (red). The positions of the amino acids are indicated. Each cloned fragment and the contained viral proteins (C, E, E2, p7, NS2, and NS3) and their corresponding amino acid positions are indicated.

**Figure 4 viruses-15-00634-f004:**
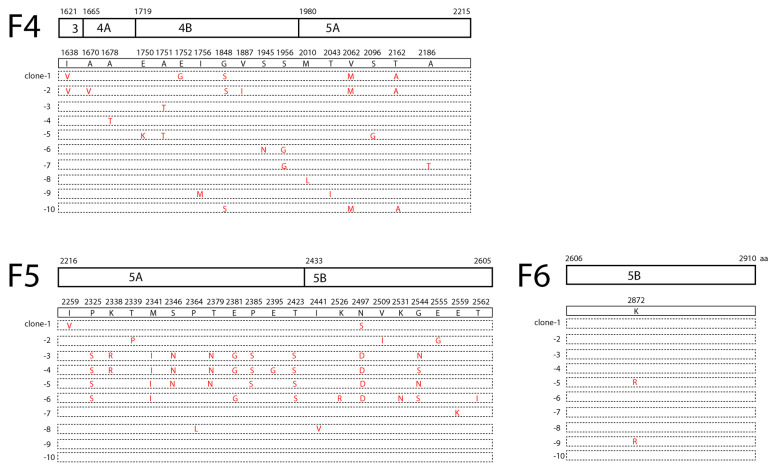
Sequence profile of HCV genome fragments−II. As described above, the amplified fragments were cloned and sequenced, and each cloned fragment was sequenced with at least ten colonies. Each dashed box indicates a sequenced cloned fragment. The amino acids different from the consensus sequence in each sequenced clone are indicated (red). The positions of the amino acids are indicated. Each cloned fragment and the contained viral proteins (NS3, 4A, 4B, 5A, 5B) and their corresponding amino acid positions are indicated.

**Figure 5 viruses-15-00634-f005:**
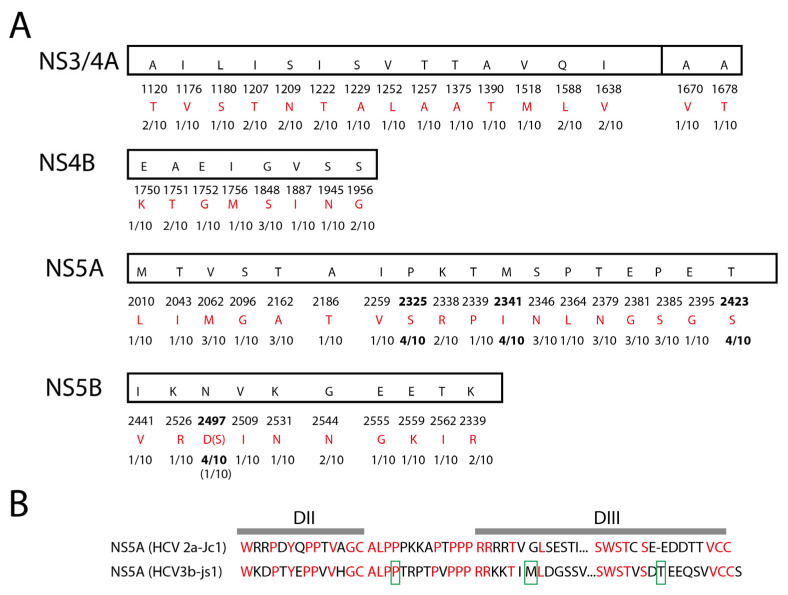
Amino acid variations of the viral non-structural proteins. (**A**) Summary of the amino acid variations of the viral non-structural proteins. The consensus sequences are shown in the boxes, and the positions of the amino acids are shown below the boxes. The variations of each amino acid (in red) and the ratio of the variations are shown as indicated. Variations with a ratio above 3/10 are in bold. (**B**) Alignment of HCV 3b-js1 NS5A with HCV 2a Jc1 NS5A. Domain II (DII) and Domain III (DIII) are indicated. The hotspot variations in HCV 3b-js1 NS5A are boxed in green.

**Figure 6 viruses-15-00634-f006:**
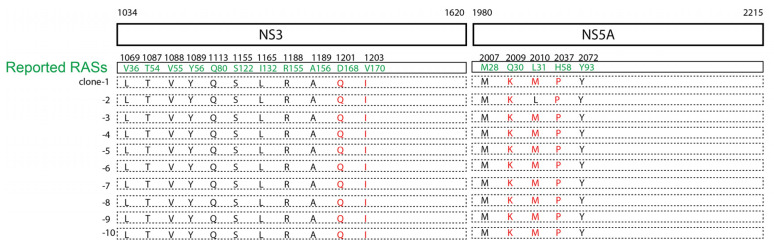
Resistance-associated substitutions (RASs) in the viral genome. The reported resistance-associated substitutions (RASs) are shown in the boxes, and their amino acid positions are shown above the boxes (green). The sequenced viral genomes are shown in the dashed boxes. The same amino acids as the reported RASs are shown in red.

**Table 1 viruses-15-00634-t001:** Primers used in this study.

Fragment	Primer	Sequence (5′–3′)	Product
F1	Sense	*ACCTGCCTCTTTCGAGGCGACACTCCACC*	1.4 kb
	Antisense	*GCCATYACGCCCCAGTGGGC*	
F2	Sense	*GCCCACTGGGGCGTRATGGC*	1.3 kb
	Antisense	*AGCTTCCCCCGRATGTGCCA*	
F3	Sense	*TGGCACATYCGGGGGAAGCT*	2.5 kb
	Antisense	*GCGTATGAGACATTTCCACATCTCATCCC*	
F4	Sense	*GGGATGAGATGTGGAAATGTCTCATACGC*	1.8 kb
	Antisense	*GGAGCYGAGAGCTGGCTGGCA*	
F5	Sense	*TGCCAGCCAGCTCTCRGCTCC*	1.1 kb
	Antisense	*AGGGCGCGTTTCTCACAGAC*	
F6	Sense	*GTCTGTGAGAAACGCGCCCT*	900 bp
	Antisense	*GCGAGAACGCGCTCAAACCATGGA*	

**Table 2 viruses-15-00634-t002:** Lymphocyte numbers and lipoproteins in the patient.

Date	CD3 Cell (690–2540 cells/μL)	CD8 Cell (190–1140 cells/μL)	CD4 Cell (410–1590 cells/μL)	CD45 Cell (900–3500 cells/μL)	HDL (1.04–1.55 mmol/L)	LDL (2.59–4.11 mmol/L)
4 June 2018	315	118	169	551	0.32	3.04
15 August 2018	327	144	127	349	0.22	1.31
12 November 2018	NA	NA	NA	NA	0.35	1.02

NA: not available.

## Data Availability

Publicly available datasets were analyzed in this study. This data can be found here: https://www.ncbi.nlm.nih.gov (accessed on 21 February 2023). The data presented in this study are available on request from the corresponding author to this end and will be openly available after upload.

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
