# Peer review of "High-Resolution Genomic Profiling of a Genotype 3b Hepatitis C Virus from a Flare of an Occult Hepatitis Patient with Acute-on-Chronic Liver Failure"

_viruses, 2023, doi:10.3390/v15030634_

Round 1

Reviewer 1 Report (Previous Reviewer 3)

I am happy with the changes the Authors have made. They have addressed all my concerns and comments.

Author Response

Thank you for your comments!

Reviewer 2 Report (Previous Reviewer 2)

Approved

Author Response

Thank you for your comments!

Reviewer 3 Report (Previous Reviewer 1)

Authors significantly improved the manuscript, and replied to almost all comments and questions posted by reviewers. However, authors have not included the limitations of the study, as it was recommended.

Also, some typos and bad wording such as “HCV RNA was detected to be lower than the limit of detection” (line 206) should be fixed.

Author Response

Thank you for the suggestions. We included limitations of this study in the modified discussion section. We went through the manuscript and did essential modifications of the typos and bad wording.

This manuscript is a resubmission of an earlier submission. The following is a list of the peer review reports and author responses from that submission.

Round 1

Reviewer 1 Report

Although authors addressed the minor comments in revised manuscript, the major concern remained unmet. The results presented in this paper still do not provide conclusive evidence in favor of occult HCV infection rather than de novo infection. Moreover, the sequencing of ten clones cannot be referred as a high-resolution genomic profiling, as indicated in the title of the paper. NGS analysis is needed to address this methodological issue. Thus, the conclusions made by authors on occult nature of HCV infection remain arguable.

Reviewer 2 Report

Yi et al is reporting an ACLF case with a flare of occult HCV infection. Upon on admission, this patient was negative for viral RNA but positive for anti-HCV antibody which needs further explanation in the discussion section (line 315). Also, the authors should include the limitations of the study.

The submitted draft has corrections already so I am assuming the authors have received some feedback. Overall, the manuscript needs thorough editing.

The authors should include the manufacturer details that include the city and country of each product [e.g. QIAgen, DNAstar ..etc]. Also, all used assays should be referenced including the ‘domestic’ assay that was used for diagnostics (lane 85). 

Reviewer 3 Report

I think there is overall merit of this manuscript to experts working in the field of Hepatitis C virus.

However, I also think the manuscript can be improved. Firstly, Figures 2 to 5, in my opinion contain too much data. I understand the Authors have done a large body of work to obtain sequence data and compare sequences. I suggest to work on these figures and try to extract the most common mutations, then only present those. This will make the figures more appealing to the reader. The tree does also contain very much detailed writing, which I think most readers will not take in. Please try to reduce the complexity and put viruses in groups, so that only the main trends are conveyed in the figure.

The authors mention immunological data with T cells. I think this reduction in T cell count is important and contribute to the flare observed in HCV infection. Please make figure which include this data. I think that will be relevant.

If blood samples are still available, please analyse lipoproteins, HDL, VLDL and LDL over the period of hospitalization and treatment. HCV associates with VLDL and it will be interesting to know if these lipoproteins were altered in size, concentration and HCV association during the cause of infection.

Thank you,